# Dietary Management of Chronic Kidney Disease and Secondary Hyperoxaluria in Patients with Short Bowel Syndrome and Type 3 Intestinal Failure

**DOI:** 10.3390/nu14081646

**Published:** 2022-04-14

**Authors:** Maciej Adler, Ewen C. Millar, Kevin A. Deans, Massimo Torreggiani, Francesca Moroni

**Affiliations:** 1Digestive Disease Unit, Aberdeen Royal Infirmary, Aberdeen AB25 2ZN, UK; maciej.adler@nhs.scot; 2Department of Applied Health Sciences, University of Aberdeen, Aberdeen AB25 2ZD, UK; 3Clinical Biochemistry, Aberdeen Royal Infirmary, Aberdeen AB25 2ZN, UK; ewen.millar@nhs.scot (E.C.M.); kevin.deans@nhs.scot (K.A.D.); 4Néphrologie et Dialyse, Centre Hospitalier Le Mans, 72037 Le Mans, France; maxtorreggiani@hotmail.com

**Keywords:** intestinal failure, hyperoxaluria, renal failure, parenteral nutrition, Crohn’s disease

## Abstract

Short gut syndrome can lead to type 3 intestinal failure, and nutrition and hydration can only be achieved with parenteral nutrition (PN). While this is a lifesaving intervention, it carries short- and long-term complications leading to complex comorbidities, including chronic kidney disease. Through a patient with devastating inflammatory bowel disease’s journey, this review article illustrates the effect of short gut and PN on kidney function, focusing on secondary hyperoxaluria and acute precipitants of glomerular filtration. In extensive small bowel resections colon in continuity promotes fluid reabsorption and hydration but predisposes to hyperoxaluria and stone disease through the impaired gut permeability and fat absorption. It is fundamental, therefore, for dietary intervention to maintain nutrition and prevent clinical deterioration (i.e., sarcopenia) but also to limit the progression of renal stone disease. Adaptation of both enteral and parenteral nutrition needs to be individualised, keeping in consideration not only patient comorbidities (short gut and jejunostomy, cirrhosis secondary to PN) but also patients’ wishes and lifestyle. A balanced multidisciplinary team (renal physician, gastroenterologist, dietician, clinical biochemist, pharmacist, etc.) plays a core role in managing complex patients, such as the one described in this review, to improve care and overall outcomes.

## 1. Introduction

Patients with type 3 (chronic) intestinal failure present unique challenges to experienced clinicians. The multi-organ involvement of the disease requires coordinated input across various medical, surgical and allied health specialities to achieve the adequate provision of nutrition, prevent and manage complications, and maintain stability and quality of life. In this paper, the authors present a case vignette of intestinal failure secondary to operative management of Crohn’s disease, complicated by enteric hyperoxaluria, nephrocalcinosis, and parenteral nutrition-associated liver cirrhosis with portal hypertension.

We analyse speciality-focussed considerations in the management of each issue, illustrate areas of conflicting evidence, and highlight the importance of the wider multidisciplinary team in patient-centred care.

## 2. The Patient’s Journey

### 2.1. Gastrointestinal

On 14 August 1996, a 29-year-old man was admitted as an emergency with an acute abdomen. His medical history included recently diagnosed Crohn’s disease with jejunal stricturing. He had developed a jejunal perforation and required extensive resection due to intraoperative findings of non-viable bowel and a strong suspicion of superior mesenteric ischemia. He had 156 cm and 17 cm of small and large bowel resected, respectively, from the jejunum onwards, leaving only an estimated 50–60 cm of the small bowel.

Over the course of the following ten years, he suffered further flares of Crohn’s disease complicated by intermittent bowel obstruction and fistulating perianal disease. Ultimately, a colostomy was created because of refractory perianal disease with pelvic abscess formation and sepsis.

Due to his young age and the implications of home parenteral nutrition (PN) on quality of life, as well as the potential long-term complications of home PN, efforts were made to maintain enteral nutrition as long as possible. Since his initial diagnosis in 1996, this was achieved through night-time nasogastric tube feeding.

By 2011 however, the patient had recurrent micronutrient deficiencies (including Vitamin A, Vitamin D, Vitamin E, Copper and Zinc) with increasing difficulties in maintaining muscle mass and body weight. MRI imaging also showed bilateral avascular necrosis of the femoral heads. A joint decision between the patient, physicians, specialist dietitians and clinical biochemistry was made to begin home PN. He remained in remission on azathioprine (1.5 mg/kg once daily) and mesalazine (500 mg three times daily) until all maintenance therapy was stopped in 2013.

After several years on total parenteral nutrition (TPN), a gradual progressive derangement of the patient’s liver function tests was noted in 2016, along with ultrasound findings suggestive of mild splenomegaly. Liver biopsy at the time was consistent with mild to moderate fibrosis, in keeping with parenteral nutrition-associated liver disease. By 2020 he was diagnosed as having liver cirrhosis with portal hypertension through a combination of clinical, biochemical, radiological, and endoscopic findings. Currently, the patient is on carvedilol for oesophageal varices and remains in a state of compensated liver cirrhosis.

### 2.2. Renal and Urinary Tract

In 2005, nine years after his Crohn’s disease diagnosis and the development of short bowel syndrome secondary to surgical resection, he began to complain of intermittent renal colic. He underwent two treatments with extracorporeal shock wave lithotripsy for right-sided obstructing ureteric calculi. In 2008, a right-sided staghorn calculus was discovered, which was treated with percutaneous nephrolithotomy. Analysis showed a hard white stone with an amorphous granular appearance composed of calcium oxalate. Biochemistry showed increased urine oxalate (0.52 mmol/L) with normal urine calcium.

Three months later, imaging suggested the formation of new stones complicated by an intrarenal abscess. Though nephrectomy was considered at the time, he successfully underwent repeated percutaneous nephrolithotomy. A renogram indicated the residual function of the right kidney was 27%, secondary to combined insults from calculi, infection, and obstruction. His renal function stabilized with creatinine levels of 150 μmol/L (eGFR 40–45 mL/min/1.73 m^2^) but did not improve following treatment.

Over the next decade, further insults included two admissions to the intensive care unit with septic shock secondary to an indwelling central venous catheter and gastroenteritis causing profound dehydration. He required hemofiltration during the first admission. His chronic kidney disease gradually progressed to stage G3b (Figure 1) with complete loss of right kidney function. He was commenced on sodium bicarbonate in 2016 for acidosis. Serial imaging revealed new developing stones in the left kidney, and by late 2017 he had multiple calculi in his left renal tract complicated by obstructive nephropathy requiring acute dialysis. The obstruction was resolved over the course of three-phased laser fragmentations.

In an effort to reduce his future stone burden, he started on calcium acetate. His TPN was adjusted to increase intravenous bicarbonate supplementation. Iron-deficient anaemia is predominantly driven by chronic low-grade gastrointestinal blood loss from the colostomy site and is managed with intravenous iron injections and blood transfusions His chronic kidney disease remains stable at stage G4 with a glomerular filtration rate (CKD-EPI) of about 28 mL/min/1.73m^2^. His blood pressure is well controlled. 

Despite these issues, the patient remains very independent and has a good quality of life. He leads an active lifestyle, working full time in farming. He has now agreed to be referred for consideration for an intestinal and multivisceral transplant.

## 3. Oxalate Nephrolithiasis and the Gastrointestinal Tract

Renal stone disease is commonly encountered in clinical practice, with a lifetime risk of up to 8.8% [1]. Calcium-containing stones account for 75% of all cases [2]. Population-wide observation studies have shown temporal variations in the incidence of nephrolithiasis, with most studies highlighting increasing disease burden with time [3,4]. Climate change, rising obesity and dietary changes have been proposed to explain this trend [5]; however, familial clustering and twin studies have also highlighted inherited predisposition to renal calculi [6]. This suggests calcium stone disease occurs through complex interactions between environmental factors and genetic susceptibility [7].

The most commonly encountered stones are comprised of calcium oxalate. Hyperoxaluria leads to urinary supersaturation of calcium oxalate and crystal precipitation, resulting in urolithiasis and nephropathy [8]. It can be either primary (inherited disorders of glyoxylate metabolism) or secondary (increased intestinal oxalate absorption). Though in the majority of cases, an underlying cause is never identified, risk factors are known to include hyperparathyroidism, renal tubular acidosis, inflammatory bowel disease, and malabsorption disorders affecting oxalate metabolism or the gut microbiota [2].

Plasma oxalate does not appear to have any physiological function and is freely filtered by the glomerulus [8]. Body oxalate originates from both dietary intake (80–130 mg/day) and endogenous hepatic production (15–45 mg/day). In health, about 5–15% of dietary oxalate is absorbed via anion exchangers, with the remainder bound to calcium and excreted in stool [2]. A diet adequate in calcium, and the presence of oxalate degrading colonic bacteria, help prevent oxalate hyperabsorption. Dehydration, hypercalciuria, dietary protein and dietary sodium contribute to a stone-forming environment.

Digestive diseases which result in fat malabsorption have been linked to enteric hyperoxaluria. These would include inflammatory bowel disease, chronic biliary and pancreatic pathologies, short bowel syndrome and gastric bypass surgery. This is in part due to a net increase in fat delivery to the colon, where it is able to bind dietary calcium, resulting in increased free oxalate, which is then absorbed into the bloodstream [8]. The presence of free fatty acids and bile salts in the colon (Figure 2) also increases the permeability of the colonic membrane [9].

The absorption of dietary oxalate depends not only on the amounts ingested but also on its bioavailability and gastrointestinal permeability, which are very different in enteral disease states than in health. Enteric hyperoxaluria was first described in the 1970s in patients who underwent surgical resection of their ileum [10,11], with findings of calcium oxalate nephrolithiasis and increased urinary excretion of oxalate postoperatively. Studies using radiolabeled oxalate showed that patients with ileal resections and hyperoxaluria absorbed five times as much oxalate as controls, eliminated by treatment with low oxalate diets [12]. The extent of ileal resection, and therefore fat malabsorption, correlated positively with the degree of pathological oxalate absorption [13]. The colon appears to be primarily responsible for this pathological increase in oxalate absorption, and colonic preservation is required for enteric hyperoxaluria [14,15,16]. Therefore, in short gut syndrome, the presence of a colon in continuity would be predisposed to nephrocalcinosis.

Human gut microflora has emerged as a key contributor to oxalate nephrolithiasis. The Gram-negative anaerobic bacterium *Oxalobacter formigenes* utilizes oxalate as a primary energy source and has been found to colonize the gastrointestinal tract in a third of individuals [17]. Case-control studies have suggested that colonization with *O. formigenes* can reduce individual risk of recurrent oxalate stone formation by up to 70% [18]. A 2018 metagenomics study compared the microbiota profiles between patients with recurrent calcium stone formation (*n* = 52) and healthy controls (*n* = 48). The authors concluded that stone-forming patients had a significantly lower bacterial representation of genes involved in oxalate degradation, inversely correlating with levels of 24 h oxalate excretion [19]. There were no differences in *Oxalobacter* abundance between groups, but *Faecalibacterium, Enterobacter* and *Dorea* were significantly lower in stone-forming patients. This suggests the role of oxalate-degrading bacteria may not be limited to the microflora of a single genus, consistent with the findings of a recent systematic review [20] and observational evidence linking recent antibiotic use with an increased risk of nephrolithiasis [21].

This raises the possibility that modification of the gut microbiome through diet or probiotics could affect urinary oxalate levels. A small placebo-controlled randomized trial found no improvement in urinary oxalate with probiotic preparations but identified improvements with a controlled low oxalate diet [22]. In patients with monogenic primary hyperoxaluria, randomized controlled trials have yet to demonstrate a beneficial effect of *O. formigenes* probiotic supplementation [23,24]; however, further studies are ongoing [25], and the intestinal environment appears to hold promise as a therapeutic target in nephrolithiasis.

Due to paucity of evidence, there are currently no treatments specific for enteric hyperoxaluria, and dietary interventions remain the cornerstone of treatment. Management of the underlying condition is also key to reducing inflammation and fat malabsorption. Oral administration of conjugated bile acid replacement was found to reduce faecal fat excretion and urinary hyperoxaluria in patients with short bowel syndrome [26] and post-surgical resection of the ileum [27] and may be considered for some patients. Preliminary data from phase I and II studies have suggested a role for oxalate specific degrading enzyme therapy in patients with secondary hyperoxaluria, and phase III trials are currently ongoing [28,29].

## 4. Crohn’s Disease, Intestinal Failure and Parenteral Nutrition Implications

Crohn’s disease is characterized by mucosal ulceration strictures and fistulae (which can be entero-enteric, entero-colic, entero-cutaneous or into other pelvic organs). It can affect all parts of the gastrointestinal tract and commonly involves the small bowel. This can lead to weight loss, malabsorption, micronutrient deficiencies, and macronutrient deficiencies [30].

Weight loss and/or macronutrient deficiency can be a consequence of disease activity and of a systemic inflammatory response. Both involve a mismatch of energy intake (loss of appetite) and expenditure (a catabolic state) and/or intestinal failure. Micronutrient deficiencies in Crohn’s can occur both because of malabsorption or because of losses (for example, iron deficiency may be due to ongoing inflammation and bleeding) [31].

According to published literature, three out of four patients with Crohn’s disease will undergo surgery during their life span [32]. Extensive small bowel resections that result in short bowel syndrome and jejunostomy are avoided if at all possible, but these are known complications of the underlying surgical management of Crohn’s [33]. Short bowel syndrome is defined as a remaining small bowel less than 200 cm [34]; if the length is under 100 cm, the patient will almost certainly require significant nutritional support, including PN.

Jejunostomy and short bowel syndrome predispose to excessive water and electrolyte losses, clinically presenting as high stoma output [35]. The small bowel has micro and macronutrient absorption functions. Water, instead, can only be absorbed within the ileum. If the Na content in the jejunum is inferior to about 120 mmol/L, there is a net movement of Na from the mucosal enterocytes into the gut lumen with a corresponding movement of water, leading to water and electrolyte losses [36]. Patients with high output stomas will often feel thirsty secondary to the loss of fluid and electrolytes, but continued oral intake of fluids can lead to further osmotic shifts that drive stoma output, leading to a vicious cycle.

In order to minimize this volume loss through a jejunostomy, glucose-sodium oral solutions can be taken to drive water across the gut mucosa, and antimotility (loperamide) or antisecretory (octreotide) drugs can be adopted. These interventions may reduce water and electrolyte losses, but they do not improve intestinal absorption. In the event of intestinal failure, they are not sufficient to support nutrition [34].

In intestinal failure, the absorptive function of the gut may not meet the minimum capacity needed to maintain macronutrients and/or water and electrolytes necessary for the survival of the individual [37]. In Crohn’s, this may be due to the disease itself or as a consequence of small bowel resections.

Intestinal failure is classified into three types:

Type 1—acute and short-lasting.

Type 2—acute but requiring nutritional support for weeks or months.

Type 3—chronic. Nutritional support in patients with intestinal failure occurs via parenteral nutrition (PN).

The patient can be supplemented by enteral nutrition (both artificial or by oral diet), but if the remaining functional jejunum is <75 cm in length, it is very unlikely for enteral nutrition to be sufficient in maintaining a nutritional state [34].

Intestinal failure represents the severest end of the spectrum of disease progression that can occur secondary to Crohn’s disease and the surgical interventions that are sometimes necessary. Patients with intestinal failure show signs of declining renal function over time, independent of other risk factors [38]. Moreover, patients with type 3 intestinal failure on home long term PN have an increased risk of developing chronic kidney disease (CKD), with a decline in creatinine clearance of about 3.5% every year and worsening kidney function in about 50% of individuals (39).

Risk factors associated with CKD development in this population are: old age, urological complications and short gut syndrome. The number of intravenous-catheter-related infections has also been identified as a risk factor for renal impairment in PN patients, as well as the volume of PN: the smaller the PN volume, the higher the risk [39,40].

Home PN (HPN) has associated risks that will impact the nutritional state of the patient and their comorbidities. Commonly, patients on PN have a high risk of line-associated sepsis and thrombosis. This not only compromises the ability to administer fluid and nutrition, resulting in dehydration, but leads to acute illness, hospitalization and, not infrequently, the need for cardiocirculatory support [41]. All of this has a significant impact on renal function in both the short and long term.

Acutely, PN patients can develop pre-renal acute kidney injury (AKI), often secondary to sepsis or hypovolemia secondary to the short gut. AKI is a predisposing factor to the non-reversible deterioration of glomerular filtration and chronic kidney disease (CKD) [42]. In patients dependent on PN, it is likely for such events to re-occur during their life, leading to slow and steady worsening of CKD secondary to repeated renal insult [43].

Long term PN is linked with other complications on the bone–kidney axis. Osteopenia and osteomalacia can both be found in patients on long term PN and lead to fractures and kidney stones [44]. Metabolic bone disease in PN subjects is multifactorial: a relative deficiency of Calcium and Phosphorus in PN can cause bone demineralization, low plasma Vitamin D concentration, and loss of post-prandial stimuli of calcitonin (via gastrin secretion) can stimulate Parathyroid Hormone, which increases bone reabsorption [45].

There are numerous hepatobiliary complications associated with long term PN use. Cholelithiasis is common in PN patients, driven by the lack of enterohepatic circulation of bile salts. Hepatic bile synthesis is regulated by the Farsenoid X Receptor (FXR) expressed in gut epithelial cells. In PN, the FXR activation is reduced due to a lack of intraluminal nutrients. Activation of FXR stimulates the expression of growth factor peptide fibroblast growth factor-19 (FGF19) [46]. FGF19 activation protects from hepatic steatosis induced by dietary fats [47].

Chronic liver disease is reported in about 25% of patients on HPN, the commonest cause being hepatosteatosis (so-called “intestinal failure associated liver disease”) [48]. This condition can be rapidly progressing, it is a cause of cirrhosis and liver impairment, and it is more common in the case of the ultrashort gut, lack of in-continuity colon and high energy and fat feeds. Cirrhosis itself has an impact on kidney function due to hemodynamic changes related to portal hypertension with arterial sequestration and reduced kidney perfusion [49,50]. Liver impairment may predispose to acute kidney injury and hepatorenal syndrome if acute stress (infection, dehydration, etc.) occurs; such events are associated with a 50% mortality [51]. Patients with cirrhosis may develop other acute complications such as gastrointestinal bleeds, sepsis and encephalopathy, which can cause direct or indirect renal impairment [44].

The beneficial role of nutrition in improving outcomes in cirrhosis is well known. Cirrhotic patients have a high risk of energy and micronutrient malnutrition, and this increases with the severity of the disease [52,53]. This is a consequence of imbalanced protein synthesis and expenditure, causing muscle and tissue wasting, reduced oral intake and increased energy expenditure because of chronic illness. There is a recognized role of hyperammonemia in the development of malnutrition and sarcopenia (that is, the loss of muscle and function secondary to malnutrition) in cirrhosis [54]. A high protein diet or high protein artificial nutrition is recommended to avoid profound sarcopenia (the recommended amount of protein is 1.5 g/kg^−1^/d^−1^) [53].

## 5. Nutrition in CKD

The relationship between parenteral support and chronic kidney disease is complex and necessitates a brief detour into general nutritional support in CKD to lay the groundwork. Similar to many other chronic conditions, protein and energy-wasting are prevalent in the end stages of renal disease [55]. Thus, it is common to manage patients with advanced CKD with nutritional supplements to avoid sarcopenia. Sarcopenia in CKD is not just a mismatch between protein breakdown and synthesis in muscles but also a consequence of high levels of uremic toxins, parathyroid hormone, angiotensin II and glucocorticoids, which all contribute to muscle atrophy [56,57]. The need to implement protein and energy intake in advanced kidney disease has to be balanced with the necessity to avoid high protein intake. High protein diets (>1.2 g/kg/day) induce an increase in intraglomerular pressure leading to hyperfiltration, secondary sclerosis and, finally, worsening of CKD. Low protein diets (LPD = 0.6–0.8 g/kg/day) not only reduce the risk of CKD worsening by blunting the effects described above but also contribute to reducing uremic toxins, phosphorus and acid intake and PTH rise, providing an overall metabolic benefit [58]. Attention to bicarbonate to correct acidosis, uric acid, and phosphorus are essential parts of nutrition interventions in CKD and end-stage renal disease [59].

Renal low protein diet (LPD) feed (Nepro^®^ LP) contains around 4 g of protein and 60 mg of sodium per 100 mL of feed, while high energy standard feeds contain around 6 g and 130 mg of protein and sodium, respectively, per 100 mL of feed. The enteral route should always be favoured when there is a functioning gut. Data from the Modified Diet in Renal Disease (MDRD) study (the largest randomized controlled trial evaluating the role of LPD in the progression of CKD) showed a potentially deleterious effect in lowering proteins in patients with renal disease as this led to unfavourable anthropomorphic measures and malnutrition [60]. However, there was an absolute deficit in calories in the LPD group compared to the control group in MDRD and all further data gathered since 1994 support the use of LPD in pre-dialysis patients [61]. Indeed, the latest guidelines in renal nutrition acknowledged this body of literature advising for LPD as early as CKD stage 3 [62]. It is important to review patients routinely, especially when acute illness can tip the metabolic balance towards more prominent catabolism. Hence, protein intake must be adjusted to cope with the new demands.

In patients on maintenance dialysis, a higher protein intake is advised. Dialysis itself provides the metabolic control an LPD may offer, while a high protein diet prevents the risk of sarcopenia and weight loss due to an imbalance between the intake and calorie demands, energy loss and protein waste [58].

PN can be considered an acute and short-term nutrition intervention, as in IF type 1, or a long-term measure. The early use of PN in an acute setting, such as intensive care units, in patients on renal replacement therapy has been associated with poorer outcomes; hence, it should be avoided [63]. There is often less fidelity in adjusting micro- and macro-nutrients in Hospital PN (this will vary from centre to centre and be dependent on the provider of the PN bag). Even though there is more fidelity in the construction of bags for Home PN patients, given the increasing BMI of the population, it can sometimes be difficult to match protein, energy and fluid requirements. PN invariably provides the majority of calories from fat or glucose; therefore, the LPD adjustment discussed above is less relevant.

Specifically for patients on dialysis, intradialytic PN can be considered to supplement nutrition [64]. However, in order to be effective, an intradialytic PN thrice-weekly should be timely started, anticipating the appearance of major alterations in nutritional markers. Since this is not always feasible in clinical practice, this approach is not widely adopted. Moreover, the majority of patients on long term PN requiring dialysis may wish to continue their routine administration of PN at home.

Patients on long term PN undergo routine nutritional screening and often require supplementation of specific micronutrients (i.e., vitamins and essential minerals). Specific attention to sodium, phosphate, bicarbonate, etc., is required in patients with CKD. These recommendations are also valid for patients receiving artificial enteral nutrition.

In exceptional circumstances such as calcium-oxalate stones uropathy, dietary adjustments are important to prevent further stone formation. Raw and cooked spinaches contain the highest amount of oxalate among foods and should be avoided in patients at high risk. The same recommendation should be taken for rhubarb and chard, while potatoes, bran, nuts and beets should be taken in moderation. It is also recommended to ingest calcium at every meal [65] and consider a low-fat diet to reduce fat excretion [66]. Colonic fat binds to intraluminal calcium, which makes it less available to combine to oxalate, thus increasing the absorption of the latter by the gut. Bile acid binders could be used to reduce fat excretion, and they have been shown to reduce oxaluria. However, they bind fat-soluble vitamins and any other oral compound if taken close to ingestion [67]. It has been shown that oral nutritional supplements and enteral feeding formulas contain a variable level of oxalate [68]. Some preparations report a potential total ingested dose of 360 mg for 24 h. Assuming that only 30% of the ingested oxalate is absorbed, this would provide >100 mg of daily oxalate, which is considered high [68]. Due to the presence of available calcium and magnesium in these feeding preparations, the bioavailability of oxalate may be reduced, lowering the risk of kidney stone formation [68]. However, it would be important for subjects at high risk to select carefully oral nutrition formulas based on oxalate levels.

Dehydration is a common issue in short gut syndrome and predisposes both to uric acid and calcium oxalate stones. Among others, the role of PN, in this case, is to increase hydration. Specific attention to high sugar levels in PN-receiving patients should be taken, and hyperglycemia should be managed effectively with tight PN regimes as this predisposes again to nephrolithiasis [67]. Intravenous Vitamin C increases oxaluria and, therefore, specific adjustments should be considered in the course of chronic PN [69].

## 6. The Multidisciplinary Team (MDT) Role

In view of the complexity of patients with intestinal failure, nutrition and medical management should be approached as a multidisciplinary team, including gastroenterologists with interest in nutrition, surgeons, chemical pathologists, dietitians, and pharmacists trained in PN dispensing and renal physicians (Figure 3). The care of these patients can be often fragmented, resulting in conflicting management and uncertainty for the patients. Artificial nutrition, both enteral and parenteral, is a prescribed treatment and, as such, should be carefully tailored to the patient’s characteristics and needs. Patients with advanced liver disease benefit from a high protein diet, but this can become counterproductive in patients with concomitant CKD, in whom a high protein load could promote renal function decline. Discussion among clinicians with expertise in managing underlying medical conditions and agreed management plans could improve the outcome of those pathologies. On the other hand, the involvement of specialized dietitians, the PN team and pharmacists could optimize nutrition regimens, reducing complications from suboptimal nutrition plans and improving the management of underlying diseases. Patient views and lifestyles should always be taken into consideration as these will impact adherence to nutrition interventions [70].

A literature review of PubMed and Scopus databases using the criteria “((secondary hyperoxaluria) OR (enteric hyperoxaluria)) AND ((intestinal failure) OR (short bowel) OR short gut))” identified 105 abstracts (Appendix A). After the removal of duplicates and non-English language articles, the remaining records were independently reviewed by two of the authors, identifying 17 case reports or case series. Of these, 15 looked at cases of malabsorption and hyperoxaluria associated with renal disease, but only 5 referred to patients undergoing small bowel surgery for reasons other than bariatric interventions or patients presenting with Crohn’s disease. In only one case series, patients received HPN. In all 17 case reports, despite emphasizing the complexity of the management of such individuals, and one case even discussing multivisceral transplantation, there was no mention of an MDT approach to achieve the best outcomes. ESPEN guidelines on chronic intestinal failure in adults [44], ESPEN guideline: Clinical nutrition in inflammatory bowel disease [30], NICE guideline on management of obesity with surgery [71], Clinical Practice Guideline for Nutrition in CKD [62] and American Clinical practice guidelines for the perioperative nutritional, metabolic, and nonsurgical support of the bariatric surgery patient [72], all encourage and support multidisciplinary interventions. On reviewing our patient’s journey, we appreciated the extended team’s effort, from medical and surgical specialists to dieticians, pharmacists and clinical biochemists who contributed to the patient’s positive outcome and good quality of life despite extensive medical comorbidities.

## 7. Conclusions

This case history illustrates the difficult and precarious clinical management of patients with intestinal failure, many of whom are on home PN. These patients have to deal not only with their primary pathology but also with the complications of their medical management, which can lead to multi-organ involvement and failure, including CKD, nephrolithiasis, issues with the bone–kidney axis, liver failure and intestinal failure.

In the case of this patient, the parenteral nutrition adjustments prevented further nephrocalcinosis complications, maintained their nutritional status and avoided the progression of sarcopenia. This was only possible with the contribution of the wider MDT. The patient is now under screening for a multi-organ transplant, which would not have been possible without a cohesive approach in their management.

## Figures and Tables

**Figure 1 nutrients-14-01646-f001:**
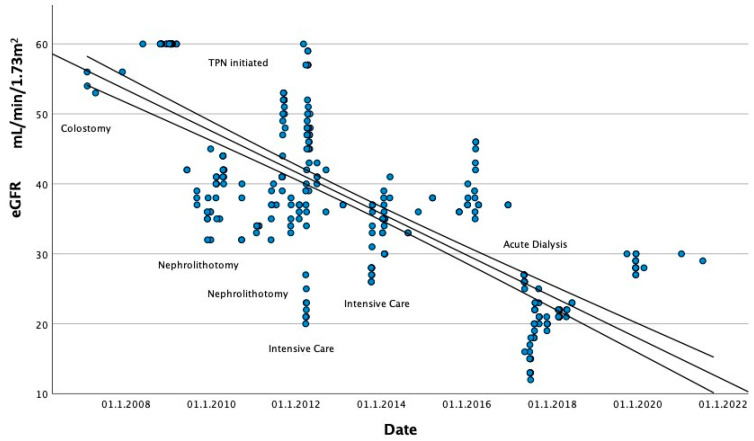
**eGFR decline over years as results of accumulation of acute insults.** This graph describes the downtrend of calculated glomerular filtration over time. The time line has been divided base on acute events to highlight the impact of specific illness or medical intervention on the subject’s renal function. It is clear that not only uropathy associated to stone disease contributed to the reduction in estimated glomerular filtration rate (eGFR) but also severe illness requiring intensive care admission and organ support had significant impact. The graph illustrates the effect of cumulative insults to the renal function.

**Figure 2 nutrients-14-01646-f002:**
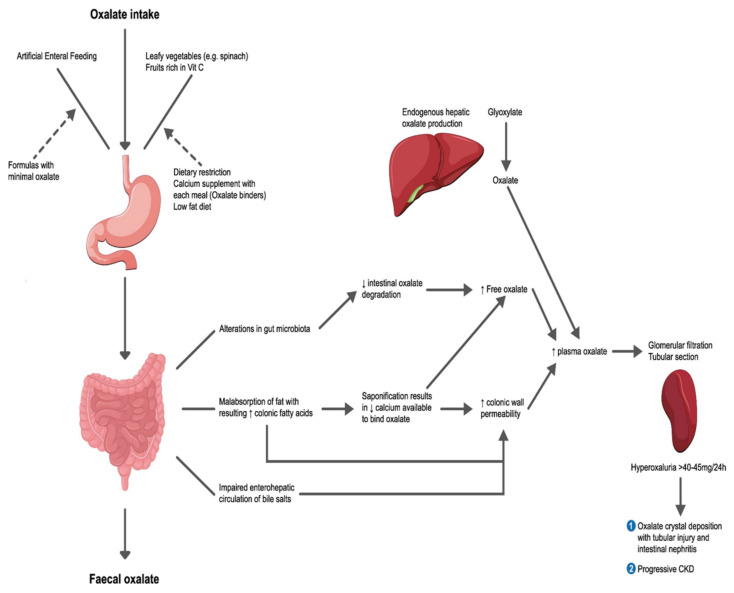
**Pathophysiology of calcium oxalate stone formation.** This image was in part adapted from Demoulin N, et al. [8] by the Medical Illustration Team at University of Aberdeen, Aberdeen, Scotland.

**Figure 3 nutrients-14-01646-f003:**
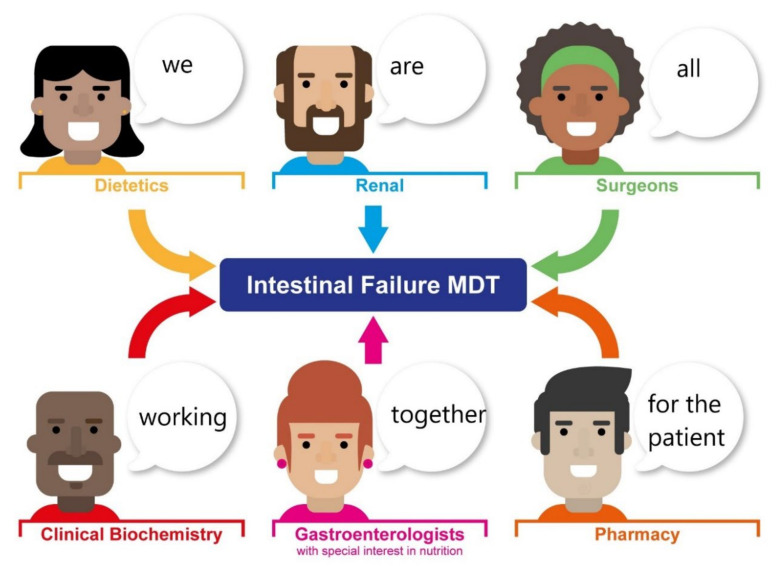
Graphical representation of multidisciplinary team meeting (MDT) managing complex patients with intestinal failure. Every member of the team plays an important role to reach the common outcome of improving patient care.

## Data Availability

Not applicable.

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
