# Peer review of "Dietary Management of Chronic Kidney Disease and Secondary Hyperoxaluria in Patients with Short Bowel Syndrome and Type 3 Intestinal Failure"

_nutrients, 2022, doi:10.3390/nu14081646_

Round 1

Reviewer 1 Report

To improve the quality of the manuscript, in my opinion, a systematic review of other clinical cases in this field is fundamental. 
if the author performed this literature review, is important to added this to the manuscript. in the actual form the revision appears like a narrative review and the case appears like a storytelling.
This is important to give clinical evidence to the presented cases.

Author Response

We thank the reviewer for the comments to our manuscript, they are pertinent and constructive.

We agree a literature review of other case reports will add value to our paper. Therefore 2 authors independently reviewed available literature and summarized findings in an added paragraph to the manuscript. It was very interesting to observe the multidisciplinary approach was not mentioned in any other case report reviewed despite been advocated in guidelines. 

We hope this addiction to our work will satisfy the reviewer.

Reviewer 2 Report

The paper looks like a case-based review, and I consider several obs

  1. The description of the case with medical details (ex. Doses, time of adm. for medications, like is azathioprine, mesalazine etc.
  2. The complete names of the authors
  3. To check the English, for ex: sarcopenia, ischemia etc (see the pdf attached).
  4. enteric vascular ischaemia (to be replaced by enteric ischemia)
  5. The 2.2 is entitled Renal..but at topic 3. the title is Oxalate nephrolithiasis..etc.
  6. Figure 1, to add more details for legend
  7. Figure 2  to add details for origin, adaptation, references
  8. for microorganisms - and microbiome in general, to write in italics the species;
  9. References for phrases as: 'In 244 patients dependent on PN it is likely for such events to re-occur during their life, leading 245 to slow and steady worsening of CKD secondary to repeated renal insult.'

Author Response

We thank the reviewer for the detailed comments and agree the suggested changes will help increasing the value of our manuscript.

  1. We added dose and administration times for medication in the case report section 
  2. Complete authors names including initial of middle names if present have been added
  3. All the British English spelling has been changed to America US English Spelling as suggested.
  4. enteric vascular ischaemia has be replaced by enteric ischemia
  5. The subtitle "Renal" in the "Patient's journey" section has been changed to "Renal and renal tract" to satisfy the nature of this patient CKD and complement the following paragraph on pathophysiology of oxalate nephrolithiasis . The title of section 3. "Oxalate nephrolithiasis and the gastrointestinal tract" has not been amended as this is a fair description of the content of the paragraph. 
  6. More details have been added to the figure 1 description.
  7. Figure 2 origin has been added in the description with reference. the figure was created by the University of Aberdeen illustration team. Contribution of medical illustration team has been acknowledgement in the figure description
  8. all name of microorganism and taxa have been changed to italic.
  9. References for phrases as: 'In patients dependent on PN it is likely for such events to re-occur during their life, leading to slow and steady worsening of CKD secondary to repeated renal insult." has been added: Pohju AK, Hakkarainen AI, Pakarinen MP, Sipponen TM. Longitudinal evolution of catheter-related bloodstream infections, kidney function and liver status in a nationwide adult intestinal failure cohort. Scand J Gastroenterol. 2022 Feb 17:1-5. doi: 10.1080/00365521.2022.2039281. Epub ahead of print. PMID: 35174757.

We think with these changes we have addressed all the comments and hope this will satisfy the reviewer. 

Round 2

Reviewer 1 Report

I suggest to insert, in a supplementary figures or in a additional figure, the Prisma diagram for the selection of the reports in literature. 

http://www.prisma-statement.org/PRISMAStatement/FlowDiagram

Author Response

We thank this reviewer for highlighting the omission of the PRISMA diagram which we have now provided in supplementary material. 

We hope this addiction to the manuscript will satisfy the reviewer. 

Reviewer 2 Report

The corrections are welcomed, I recommend accepting the manuscript.

Author Response

We thank the reviewer for accepting our corrections.